# MITIGATING THE INFLUENCE OF DISTRACTOR TASKS IN LMS WITH PRIOR-AWARE DECODING

## ABSTRACT

The broad capabilities of Language Models (LMs) can be limited by their sensitivity to distractor tasks: LMs can infer secondary tasks from the prompt in addition to the intended one, leading to unwanted outputs. For example, prompt injection attacks can cause models to deviate from explicit directives. In some 'inverse scaling' cases, this unwanted behaviour actually worsens as models scale up to at least 540B parameters. We present a theoretical framework that interprets LMs as a product of experts that combine multiple data generation processes. Based on this framework, we introduce prior-aware decoding (PAD) – a simple contrastive inference method to reduce the influence of distractor tasks. We apply PAD to eleven models, across four datasets, and find improvements in 41 out of 44 task-model combinations, with a median increase in task completion proportion of 40%. The results suggest a promising direction for further development towards more reliable language models.

## 1 INTRODUCTION

Language models (LMs) have come to occupy a central role in a wide range of tasks, from data processing to the creation of instruction-following assistants. These models seem to both increase in performance and also develop new capabilities as they scale up in parameters, especially when given examples of a task (Radford et al., 2019; Brown et al., 2020; Wei et al., 2022). They see both widespread public use, and increasing integration into sensitive tasks where their versatile capabilities are key (Javaid et al., 2022). However, this increasing reliance on LMs raises concerns about their reliability and potential vulnerabilities. Their extremely general ability to recognise patterns may come with a cost: language models are susceptible to *distractor tasks*, unintended secondary tasks which a model can infer from the prompt in addition to the intended one, leading to unwanted outputs (Wei et al., 2023).

Distractor tasks include both semantically complex cases like prompt injection, in which the distractor task of following the most recently injected instruction overrides the intended task of following the initial instructions, and much more semantically simple cases where a model defaults to outputting a common pattern regardless of prior context (Greshake et al., 2023; McKenzie et al., 2023).

Even cutting-edge models struggle with trivial tasks in the presence of very common patterns. For example, as of September 2024 GPT-4o, when prompted with "`Write all the numbers up to 30 except multiples of 13, and nothing else`", will correctly exclude 13 but fail to exclude 26. Other examples of such distractor tasks include repeating common misconceptions instead of true answers (Lin et al., 2022) and failing to write mathematical proofs when the correct proof is very similar to an incorrect but much more common proof (Collins et al., 2024).

Furthermore, for some tasks of this type, larger models actually perform worse: similar models with more parameters can be *more likely* to repeat common misconceptions and succumb to prompt injection attacks (McKenzie et al., 2023; Lin et al., 2022). This phenomenon, known as 'inverse scaling' (Lin et al., 2022), reveals a pressing need to better understand how these problems arise and how they can be fixed. Unlike other problems with LMs which are usually resolved by scaling up the model and training data size, the existing problems might become even more pronounced, and other similar issues might emerge.

## 1.1 PRIOR-AWARE DECODING

To better understand and address the problem of distractor tasks, we characterise language models as a *product of experts* (Hinton, 1999) which combine the predictions of several component models, each component model predicting the next token assuming a particular task or context. This framework, presented in detail in Section 4, allows us to decompose the model's behavior into distinct "experts" that may be responsible for different aspects of the output, which in turn lets us interpret the problem of distractor tasks as models overweighing a particular expert.

Using this framework, we present a technique to reduce the influence of distractor tasks in cases where the prompt can be divided into one part containing the intended task and one part containing the distractor task and other important information. In this case, we can contrast the outputs the model gives for the full prompt with the outputs it gives for just the part which yields the distractor task, extrapolating to infer how it would behave with less influence from the distractor task. This is a form of contrastive decoding (Li et al., 2023). Our approach differs from previous methods by specifically targeting the influence of distractor tasks and leveraging the product of experts model

More specifically, our method generates the original logits $L_O$ from the original prompt, as well as the weakened logits $L_W$ from the portion of the prompt which does not contain the original task. Even though $L_O$ and $L_W$ would both, by default, favour undesired outputs consistent with the distractor task, $L_O$ gives a higher probability to outputs consistent with the intended task. So, by sampling instead from a linear combination of the logits, $L_O + \alpha(L_O - L_W)$ we can selectively favour outputs which are more consistent with the original task. ($\alpha$ here is a tunable hyperparameter.) Using this technique we can mitigate the influence of the distractor task, as we illustrate in Figure 1.

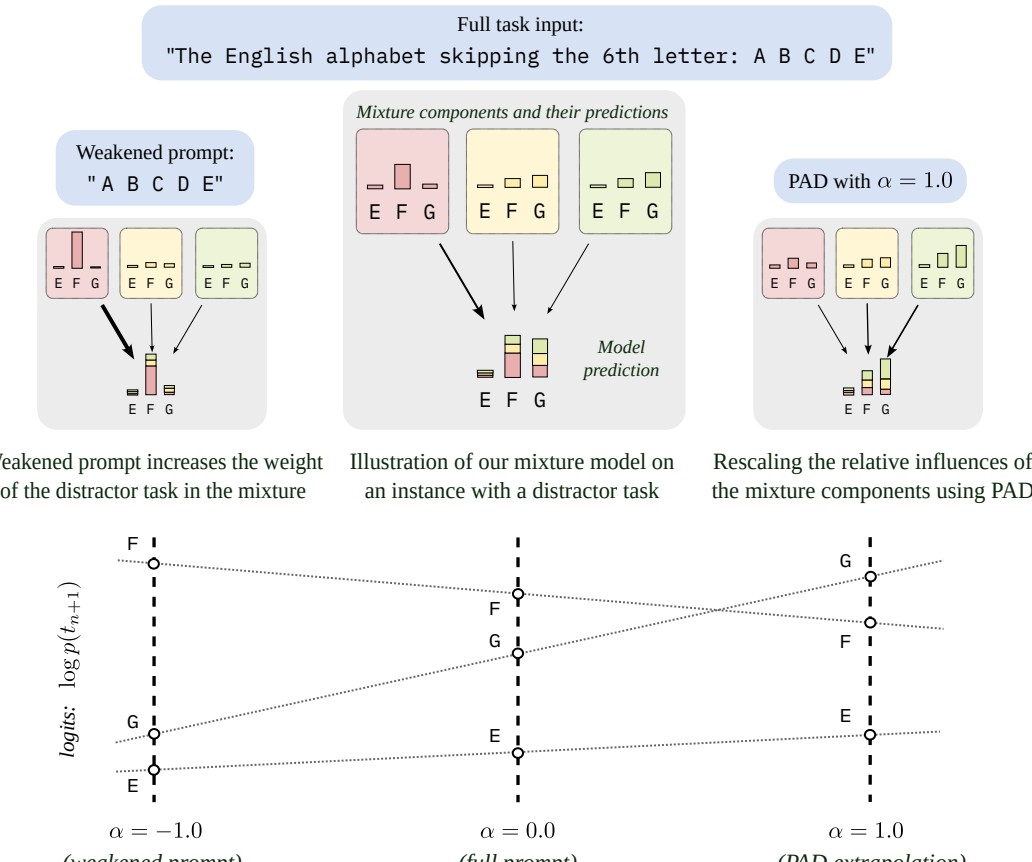

Figure 1: Illustration of our geometric mixture model, also referred to as product of experts, and Prior-Aware Decoding in a task requiring a modification of a very common sequence.

We test the technique on eleven different models including GPT-2, Llama-2, and Mistral, using four tasks to evaluate susceptibility to distractor tasks. Three of these tasks are from the Inverse Scaling Dataset (McKenzie et al., 2023; Wei et al., 2023) which exhibit strong priors, including a prompt injection task, and one task is a custom task which challenges models to produce modifications of common sequences.

Our primary baseline is the unmodified completion according to the same model. Across the 44 model-task combinations we find that our technique outperforms the baseline on 41 combinations, with a median absolute increase in proportion of tasks completed of 40%. See Figure 2 and Section 6 for the results. These improvements demonstrate the potential of our method to significantly enhance the reliability and performance of language models across a variety of model architectures in scenarios featuring distractor tasks.

The technique requires two choices: selection of the weakened prompt, and the coefficient $\alpha$. We offer consistent approaches to generating the weakened prompts for each of our datasets and also propose a general method for weakened prompt construction. We explore a range of values for $\alpha$. Crucially, this method does not require retraining: it operates at inference time by making two queries instead of one and computing a linear combination of the resulting logits.

The improvements in accuracy due to our technique suggest that viewing LMs as product of experts models can be a useful framework for understanding and improving the behaviour of these models. Our research provides new approaches for utilizing capabilities which are dormant in models and which have been hard to elicit with prompting alone. This includes capabilities of immediate practical importance, such as resistance to prompt injection attacks, as well as understanding how to correct for biases learned in the training data.

The main contributions of this paper are:

- Interpreting language models as geometric mixtures (i.e. a product of experts) to develop a model of how distractor tasks cause poor performance on certain inputs.

- Based on that model, proposing a general framework for eliciting other component of the geometric mixtures and extrapolating to more desirable mixtures in logits space.

- Developing a technique based on the product of experts model and showing that it robustly improves performance across several language models on several tasks from the Inverse Scaling dataset.

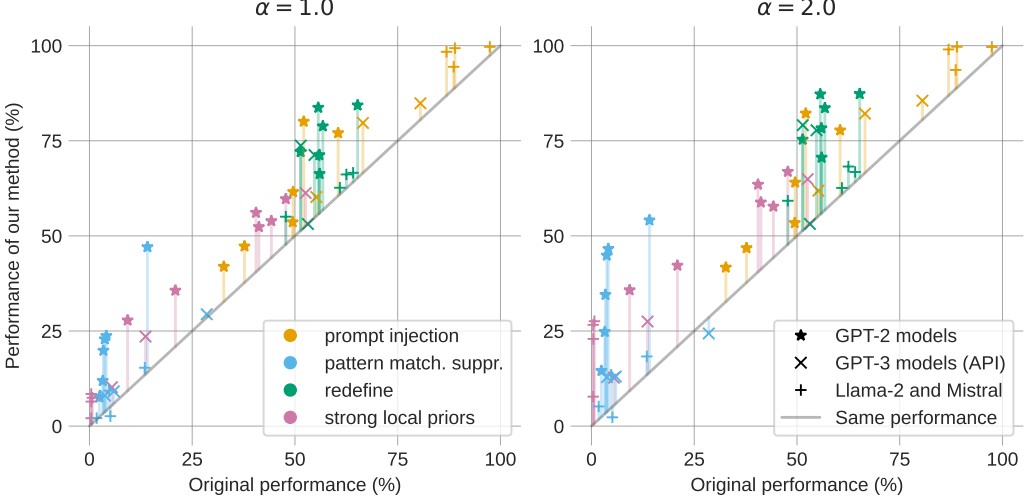

Figure 2: Overview of the results of the Prior-Aware Decoding method at different values of parameter $\alpha$ over 4 experimental task sets and 11 language models. Each point refers to performance on one (task set, model) pair, showing the baseline performance of the unmodified original model ($x$-axis) vs PAD using the "truncated prompt" weakening ($y$-axis). See Table 1 for details.

## 2 RELATED WORK

**Inverse Scaling.** For most tasks, performance scales together with the model scale (model size, dataset size, and training compute). However, certain tasks exhibit *inverse scaling*, where increasing the scale of the model leads to worse performance. The term was introduced by Lin et al. (2022) to characterise the tendency of LMs to repeat common misconceptions. McKenzie et al. (2023) further developed the notion and provided the Inverse Scaling dataset as well as several informal categories. Wei et al. (2023) adds to the discussion, showing that of the original eleven tasks in the Inverse Scaling dataset, only four persist in demonstrating inverse scaling on models of up to 540B parameters. We apply our method to three of those four tasks. Our work is the first to offer a formal account of the mechanisms that have so far only been informally hypothesised in Wei et al. (2023), and we use this to develop a novel technique that improves performance.

**Problems Related to Strong Priors.** While in this paper we investigate the problem of strong priors in the domain of language modelling (discussed primarily in Section 3), a somewhat similar problem also occurs in diffusion models for image generation. Diffusion models have been observed to some-times struggle to generate images that have a precise correspondence to the prompt. This problem is especially pronounced for complex prompts that include rare words, or describe implausible scenes, spatial relations, or compositionality. One technique for mitigating this problem in diffusion models is classifier-free guidance (Ho & Salimans, 2021). It samples the diffusion model using a linear combination of conditioned and unconditioned score estimates $(1 + w) \cdot e(z_t, c) - w \cdot e(z_t)$, where $z_t$ is the image at the denoising step $t$, $c$ is the conditioning information such as the prompt, $e$ is the model and $w$ is the guidance weight. High guidance weight improves text-image alignment (Saharia et al., 2022), although high values often lead to over-saturated images or have a generally unnatural appearance. The technique that we present in this paper uses a somewhat similar combination of differently conditioned model outputs to mitigate the problem of strong priors in LMs.

**Surface Form Competition and DC-PMI.** Holtzman et al. (2021) coin 'surface form competition' as a phenomenon where different tokens representing the same concept compete for probability. For instance, the terms "computer" and "PC" represent the same concept but are different surface forms. Since the probability mass is finite, this competition lowers the probability of the correct answer. Domain Conditional Pointwise Mutual Information (DC-PMI) addresses this issue by reweighing each option according to its likelihood within the context of a specific task, calculated by sampling from a subset of the prompt. This approach yields gains in zero-shot performance for a variety of GPT-2 and GPT-3 models on a selection of multiple-choice datasets. Our work applies a similar kind of technique - reweighing based on likelihood - but we do so to address a different set of tasks, including those which exhibit inverse scaling.

**Contrastive Inference Methods.** Contrastive Inference methods steer language models towards desired outputs by taking multiple samples across different settings and favouring outputs which are comparatively more likely in one sample than in another. The original implementation in Li et al. (2023) takes output distributions using the same prompt in two models, one smaller and one larger. It then contrasts the logits from the two models, selecting the token with the largest difference between the original ($L_O$) and smaller model ($L_S$), i.e. $\max_{t \in T'}(L_O(t) - L_S(t))$, limited to a subset $T'$ of top tokens to mitigate noise in low-likelihood tokens.

Our approach is most similar to the subsequent techniques introduced in Shi et al. (2024) and Malkin et al. (2022), which apply contrastive decoding to pairs of shorter and longer prompts, using the same formula we apply. Malkin et al. (2022) contrasts the full prompt against only the text generated so far to increase long-range semantic coherence, finding improvement in generic language understanding and knowledge retrieval tasks. Shi et al. (2024) uses an equivalent formula to handle conflicts between prior knowledge and information in the context. They also interpret the combination as a product-of-experts.

Compared to these works, ours frames the language model itself as a product-of-experts that must select between specific competing tasks, some of which are unwanted distractors, and the resulting manipulation of logits as a reweighing of the experts. Furthermore, while Malkin et al. (2022) and Shi et al. (2024) focus on models failing to account for knowledge and facts, we emphasise how they can include even extremely semantically simple patterns like sequences of numbers. We also draw the connection to inverse scaling.

## 3 STRONG PRIORS IN GENERATIVE MODELS

In this section, we introduce the concept of *strong priors* within generative models, focusing on their manifestation in language models and discussing their broader implications.

The term "strong priors" in this context was introduced by McKenzie et al. (2023) as one of the categories of LM tasks that exhibit inverse scaling trends. The difficulty of this category of tasks seems to largely stem from local context overly influencing the predicted continuation. We aim to extend the informal characterisation in McKenzie et al. (2023) with a more comprehensive definition.

**General Strong Priors.** Generative models can exhibit a phenomenon where a part of the input disproportionately affects the output, substantially diminishing the impact of other parts of the input that are essential for the model to complete its assigned task. This general notion of strong priors is applicable to generative models for various domains, including language modelling and image generation.

One example of the phenomenon is an autoregressive language model that has been prompted to write a common sequence (the alphabet) with a variation in the middle (a skipped letter). We empirically observe that once the LM has written enough of the common sequence, it fails to add the variation. We hypothesise that the model is overgeneralising from a frequent pattern in its training data, ignoring other context.

**Definition (Strong Prior).** Consider a model input $\{T, D\}$, where $T$ is the *task description part* that may include examples, and $D$ is the *data part*, consisting of the task input and possibly part of the output. Let $M$ denote a generative model, with $M$ determining any parameters of the generation process, such as temperature, and let $p_M(O|T, D)$ represent the probability of $M$ producing $O$ as its output given $T$ and $D$. Finally, let $\approx$ be a similarity relation on probability distributions. Then $D$ induces a strong prior in $M$ relative to $T$ if:

$$p_M(\,\cdot\,|T, D) \approx p_M(\,\cdot\,|D)$$

This implies that the contribution of $T$ to the model's output is insignificant, and the output is mostly determined by $D$. The result is that the model tends towards specific outputs when provided with data $D$, regardless of the task description $T$ provided.

This definition is intentionally broad, to allow adaptations based on the domain, modality, and the specific case in focus. The partitioning of the model input, or the prompt, into task $T$ and data $D$ components, is context-dependent, and may not be feasible or pertinent in every setting. Moreover, the interpretation of $p_M(\,\cdot\,|T, D) \approx p_M(\,\cdot\,|D)$ depends on the modeling assumptions, with our proposed default metric being the total variational distance $\delta$. This would imply the presence of strong priors if $\delta\left(p_M(\,\cdot\,|T, D), p_M(\,\cdot\,|D)\right) \leq \varepsilon$, where $\varepsilon > 0$ is a predetermined threshold.

While the Kullback–Leibler divergence $D_{KL}\left(p_M(\,\cdot\,|T, D)\|p_M(\,\cdot\,|D)\right)$ may be more suitable than $\delta$ in certain contexts, it is susceptible to significant influences from relatively improbable events that exhibit substantial shifts in log-probability.

**Strong priors in language models.** In this subsection, we adapt the general concept of strong priors for language models. We also explore the common case of local strong priors, where we assume that the model input is a prompt of the form $P = TD$. Additionally, we suggest using the next *token* as the main unit of output in the distribution $p_M(\cdot|P)$. Despite potential ambiguity in the separation into task description $T$ and data $D$, this representation aligns with the structure of many LM benchmarks and common prompting strategies. We can characterise strong local priors as cases where prepending the task description to the data is not significantly influencing the distribution:

$$\delta\left(p_M(\,\cdot\,|TD), p_M(\,\cdot\,|D)\right) \leq \varepsilon$$

In instances where $p_M(\cdot|TD) \approx p_M(\cdot|D)$, the difference between the two distributions may hold significant information: the correct continuation is often more probable in $p_M(\cdot|TD)$ than $p_M(\cdot|D)$. We present a novel method to compensate for strong local priors based on our model in Section 5. This intuition would predict that our method works better in the presence of strong priors, that is with smaller values of $\delta$ as introduced above. This relationship on our dataset is indeed demonstrated in Appendix C and Figure 4.

While we focus on the local case, the above foundational definitions as well as the proposed method can be adapted for non-local scenarios where $T$ and $D$ are intertwined in more complex way but still allow for a meaningful systematic distinction between task and data. We include a concrete example in Appendix A and empirical measurements of $\delta$ in Appendix F.

## 4 LANGUAGE MODELS AS PRODUCTS OF EXPERTS

In this section, we present our proposed model for addressing the challenge of distractor tasks emerging from strong priors in generative language models. Our approach models the generative distribution as a product of experts (i.e. a geometric mixture) of two components: one that captures the correct continuation of context based on the intended task, and another that captures continuation based on the distractor task induced by strong local priors. We then define a new distribution that weighs between these components to produce more contextually appropriate completions.

Consider a generative language model $M$ with its generative distribution given by $p_M(t_{n+1}|t_1 \ldots t_n)$. We assume that this distribution is influenced by two main forces:

- $p_C(t_{n+1}|t_1 \ldots t_n)$: The true continuation distribution which considers the entire context.
- $p_L(t_{n+1}|t_1 \ldots t_n)$: The distribution driven by strong local contexts or priors, which can sometimes overshadow the global context.

The distributions $p_C$ and $p_L$ may overlap if there is no strong local prior influencing the continuation.

**Geometric Mixture Model.** We assume that $p_M$ can be modeled as a geometric mixture of $p_C$ and $p_L$, also known as a Product of Experts:

$$p_M(t_{n+1}|t_1 \ldots t_n) \propto p_C(t_{n+1}|t_1 \ldots t_n)^{\gamma^*} p_L(t_{n+1}|t_1 \ldots t_n)^{1-\gamma^*}$$

where $0 \leq \gamma^* \leq 1$ is an apriori unknown parameter of the mixture.

To distinguish between contextually correct completions and those biased by strong priors, we aim to extract $p_C$ from $p_M$ and $p_L$. Using a substitution $\alpha^* = \frac{1-\gamma^*}{\gamma^*}$, we define:

$$p_C \propto p_M^{1+\alpha^*} p_L^{-\alpha^*}$$

which lets us define a generalized $p_\alpha$ for any real $\alpha$:

$$p_\alpha \propto p_M^{1+\alpha} p_L^{-\alpha} = p_M \left( \frac{p_M}{p_L} \right)^\alpha = p_L \left( \frac{p_M}{p_L} \right)^{1+\alpha}.$$

These expressions provide different interpretations of $p_\alpha$ and therefore also of $p_C = p_{\alpha^*}$: As a geometric mixture of $p_M$ and $p_L$, as a reweighing of $p_M$ based on the ratio $\frac{p_M}{p_L}$, and as a form of weighed importance sampling, interpreting $p_L$ as weighted negative evidence.

**Logits Formulation.** In most generative language models, the next token follows a Boltzmann distribution defined by:

$$p(t_{n+1}|t_1 \ldots t_n) \propto e^{l(t_{n+1}|t_1 \ldots t_n)/T}$$

where $l(t_{n+1}|t_1 \ldots t_n)$ are the logits and $T \geq 0$ is a temperature parameter.

Given this, our model can be expressed in terms of logits as $l_\alpha(t_{n+1}|t_1 \ldots t_n) = l_M + \alpha(l_M - l_L)$ leading to:

$$p_\alpha(t_{n+1}|t_1 \ldots t_n) \propto \exp \left( \frac{(1+\alpha)l_M(t_{n+1}|t_1 \ldots t_n) - \alpha l_L(t_{n+1}|t_1 \ldots t_n)}{T} \right).$$

With this formulation, our model strategically weights the logits of the generative model and the logits driven by local priors to generate more appropriate completions given strong priors.

It is crucial to highlight that $\alpha$ plays a determining role in model behavior. Large values of $\alpha > \alpha^*$ can severely impair the model's performance. This mainly stems from amplifying noise variance in model outputs, especially given potential non-specific model errors represented as $l_M = \hat{l}_M +$

$N(0, \sigma^2)$. Additionally, the deviation of model $L$ from model $M$ can be more extensive than just the influence of dominant local priors. Such deviations may inadvertently favor incorrect continuations.

**Approximating a strong prior model.** To construct a strong prior model $L$ - an approximation of model $M$ that is more prone to be influenced by strong priors - we alter the prompts to accentuate this behavior, and then employ $M$ on these modified prompts. The main modification we consider is stripping the task from the prompt, although in Appendix E we discuss a different approach based on comparing model outputs with and without a weakening system prompt that is task-agnostic.

We reduce the context available to the model by removing the task description at the start of the prompt. For a context $P$ (the task description, including few-shot examples if present) followed by $t_1 \ldots t_n$ (the task input and an initial segment of the output), our altered logits are given by:

$$l_L^{\text{strip}}(t_{n+1}|Pt_1 \ldots t_n) = l_M(t_{n+1}|t_1 \ldots t_n)$$

For instance, for the task: *"View number as text. Do not perform computation. Q: What is the first digit of 23+63?"*, $P$ represents the text: *"View number as text. Do not perform computation."* and $t_1 \ldots t_n$ captures *" Q: What is the first digit of 23+63?"*.

## 5 METHODOLOGY AND EXPERIMENTAL SETUP

We employ our proposed technique of Prior-Aware Decoding to mitigate the influence of distractor tasks. The influence of the undesired underlying prior is captured by querying the model for logits using a *weakened prompt*, which is more susceptible to the prior's influence.

The PAD technique is implemented through the following steps. For a more detailed description of the algorithm, see Appendix D.

1. For each prompt in the dataset, we generate two versions:
   - An original prompt containing both the task description and the data.
   - A weakened prompt more likely to give outputs consistent with the distractor task.
2. We query the model on both the original and weakened prompts to obtain two sets of logits.
3. We compute a linear combination of these logits: $L = L_O + \alpha(L_O - L_W)$, where $L_O$ are the original logits, $L_W$ are the weakened logits, and $\alpha$ is our extrapolation parameter.
4. We sample from this modified distribution to generate the output.
5. We calculate the average performance across the entire dataset.

For our main results we produce weakened prompts by removing the task component from the prompt. We illustrate examples of splitting prompts from each dataset in Table 2: the split was unambiguous for the tasks we selected, and generated by a regex script. We also discuss weakening prompts by prepending a system prompt in Appendix E.

### 5.1 EXPERIMENTAL SETUP

**Comparisons.** We test our method across four sets of tasks, on eleven models, at $\alpha = 0$ (the original model baseline), $\alpha = 1$, and $\alpha = 2$. These specific values were chosen as our initial exploration points, with $\alpha = 0$ serving as a control. We provide a more comprehensive analysis across a broader range of $\alpha$ values in Appendix G.

**Tasks:** We evaluate our model on four task sets: three from the Inverse Scaling Prize (*Prompt Injection*, *Pattern Match Suppression*, *Redefine*), and one custom set (*Strong Local Priors*). The Inverse Scaling Prize tasks were specifically selected for their clean task/data decomposition and their demonstrated inverse scaling behavior up to 540B parameter models, as reported by Wei et al. (2023). Table 2 provides examples of prompts from each dataset, illustrating the task/data split.

**Model Architecture.** Our experimental models comprise a range of transformer architectures, including four GPT-2 sizes, two Llama 2 sizes, two Mistral models, and several OpenAI models via API. This diverse selection allows us to compare results both between models and across different sizes of a fixed model, providing a broad view of PAD's effectiveness across various model architectures. For small local experiments we used GTX 1060-Ti, and for larger ones a rented A6000.

**Metric.** We use a *Probability of Correct Completion* as the metric to evaluate model performance. This metric naturally captures the success rate of the model in fulfilling the requirements of the task description. A completion is correct if it fulfills the requirements of the task description, and incorrect otherwise.

Our main results show models at temperature 1.0: results for temperature 0.0 are in Appendix G.

## 6 RESULTS

The comparisons are recorded in Table 1, with a graphical representation in Figure 2. Notably, extrapolating with $\alpha = 2$ outperforms the baseline in 41/44 cases, with a 30 percentage point (%pt) mean completion improvement across all tasks and models. For $\alpha = 1$, the improvement is 17.3%pt.

As Figure 2 shows, in many cases our method causes models to double the proportion of tasks they complete. This is not properly reflected by the mean: If we instead consider the number of tasks from a given dataset completed by a given model, we find the median percentage increase at $\alpha = 2$ compared to the baseline is 40%, and for $\alpha = 1$ compared to the baseline it is 27%. Figure 3 in Appendix C presents the results in terms of relative information gain.

The results also demonstrate some amount of inverse scaling within model families at all values of $\alpha$, and even in the cases where larger models do see better performance, PAD produces greater gains than doubling the number of parameters. As an additional baseline we investigate applying PAD using a smaller model from the same family as the weakened model, essentially following the approach in the original contrastive method used by Li et al. (2023), and find little gain, with only 10 out of 16 tasks-model pairs showing any improvement at $\alpha = 1.0$ (in contrast to 41 of 44 of task-model pairs of our main result), and with a median increase in task completion proportion of 0.3% (in contrast to 27% of our main result). See Appendix H for details.

Extrapolation from a system prompt generally gives worse results, although notably it outperforms the baseline in all four task/model/parameter combinations where extrapolation from the truncated prompt does not. We give more details in Appendix E.

The three cases where extrapolation to $\alpha = 2$ yield lower results (pattern matching suppression on gpt3.5-turbo instruct, pattern matching suppression on gpt2-medium, and redefine on text-ada-001) do not seem to point to any general weaknesses of the technique. Interestingly, gpt3.5-turbo-instruct and ada-001 exhibit the highest TVD on pattern matching suppression and redefine, respectively, both substantially higher than the next highest, suggesting the potential absence of a strong prior in these cases (see AppendixF for a full table of TVD values).

We explore a wider range of values of $\alpha$ in Appendix G. In particular, Figure 5 and Figure 6 show the probability of correct completion on a variety of models across a range of values of $\alpha$ from -2 to 3. We show that range to more clearly demonstrate general trends in completion as $\alpha$ changes. In general, higher values of $\alpha$ sometimes continue to produce better results, but often the benefits tail off, and in some cases the graphs show a peak beyond which higher values of $\alpha$ cause slow degradation.

## 7 CONCLUSION AND FUTURE WORK

This paper introduces Prior-Aware Decoding (PAD), a novel technique for mitigating the influence of distractor tasks in language models. Our approach is grounded in a theoretical framework that conceptualizes language models as products of experts, offering a new perspective on how distractor tasks emerge and persist.

The primary theoretical contribution of this work is the framing of language models as products of experts. This model provides a specific explanation for the emergence of distractor tasks and offers insights into why we observe counterintuitive scaling behaviors, including inverse scaling, on certain tasks. Importantly, our framework suggests that the balance between these "experts" is not necessarily dictated by model size, which could explain the lack of straightforward scaling effects and even the inverse scaling phenomena observed in some cases.

| model | prompt injection | | | pattern match. suppression | | | redefine | | | strong local priors | | |
|---|---|---|---|---|---|---|---|---|---|---|---|---|
| | $p_0$ | $p_{1.0}$ | $p_{2.0}$ | $p_0$ | $p_{1.0}$ | $p_{2.0}$ | $p_0$ | $p_{1.0}$ | $p_{2.0}$ | $p_0$ | $p_{1.0}$ | $p_{2.0}$ |
| Llama2 7B | 49.5 | 53.6 | **53.4** | 3.2 | 11.9 | **24.8** | 55.9 | 71.2 | **78.3** | 20.9 | 35.7 | **42.2** |
| Llama2 13B | 32.7 | **41.9** | 41.7 | 4.1 | 23.7 | **46.5** | 56.8 | 78.9 | **83.6** | 41.2 | 52.4 | **58.8** |
| Mistral 7B | 60.5 | 77.0 | **77.8** | 3.4 | 19.9 | **34.5** | 55.7 | 83.7 | **87.2** | 47.8 | 59.7 | **66.9** |
| Mixtral 8x7B | 52.2 | 80.2 | **82.4** | 14.1 | 46.7 | **53.8** | 65.4 | 84.7 | **87.6** | 44.3 | 53.7 | **57.7** |
| gpt2 | 88.6 | **94.4** | 93.6 | 13.5 | 15.3 | **18.3** | 62.5 | 66.1 | **68.2** | 0.4 | 2.1 | **7.7** |
| gpt2-medium | 88.9 | 99.3 | **99.7** | **5.1** | 2.6 | 2.3 | 60.9 | **62.6** | 62.6 | 0.7 | 7.5 | **27.6** |
| gpt2-large | 97.4 | **99.7** | 99.6 | 4.8 | 9.3 | **13.5** | 64.2 | 66.6 | **66.8** | 0.5 | 6.4 | **22.9** |
| gpt2-xl | 86.9 | 98.4 | **99.0** | 1.8 | 2.1 | **5.2** | 47.8 | 55.0 | **59.2** | 0.4 | 8.5 | **26.6** |
| text-ada-001 | 66.5 | 79.7 | **82.1** | 3.8 | 8.0 | **12.9** | **53.2** | 53.1 | 53.1 | 5.5 | 10.2 | **12.7** |
| davinci-002 | 55.2 | 60.2 | **61.9** | 5.9 | 9.2 | **13.0** | 54.7 | 71.3 | **77.8** | 13.6 | 23.5 | **27.5** |
| gpt3.5-t.-instr. | 80.5 | 84.8 | **85.5** | 28.5 | **29.4** | 24.4 | 51.4 | 73.7 | **79.1** | 52.6 | 61.3 | **64.9** |
| *mean* | 63.2 | 80.8 | **89.7** | 7.3 | 23.2 | **37.4** | 52.4 | 72.3 | **83.6** | 19.0 | 35.1 | **51.3** |

Table 1: Percent probabilities of correct completion for different tasks and language models. $p_0$ indicates full-prompt accuracy (baseline); $p_{1.0}$ and $p_{2.0}$ indicates using truncated prompt as the weakened prompt with parameter $\alpha = 1.0$ and $\alpha = 2.0$. Bold entries denote the highest-scoring parameter for a given task and model. All probabilities are reported at model temperature $T = 1.0$.

Our experimental results demonstrate the efficacy of PAD across a range of language models and tasks from the Inverse Scaling dataset. We consistently observed improvements in model performance, with a median increase in task completion proportion of 40% at $\alpha = 2$. These findings not only validate the practical utility of our approach but also provide empirical support for our theoretical framework.

The implications of this work extend beyond the specific technique we've developed. Our results suggest that it may be possible to elicit greater capabilities from models even in cases where straightforward scaling is counterproductive, by employing more sophisticated forms of inference, and that this may be necessary to account for certain persistent quirks.

While our current study focuses on relatively small models and a specific set of tasks, the potential applications of this work are broad. Perhaps most promisingly, PAD could be developed into a robust defense against jailbreaks and prompt injection attacks, although further research with more challenging datasets is needed to fully assess its potential in this area.

Looking forward, we identify several key directions for future research:

1. **Model Interpretability:** Investigating how well our theoretical framework aligns with the actual internal processes of language models could provide valuable insights into model behavior and further validate our approach.

2. **Extension to Harder Cases:** Applying and adapting PAD to more challenging instances of prompt injection and other distractor tasks could enhance its practical utility.

3. **More Sophisticated Inference-time Techniques:** Combining our method and underlying model with related approaches like classifier-free guidance (Ho & Salimans, 2021), and activation addition (Turner et al., 2023), to build more robust approaches for steering language models.

In conclusion, this work not only introduces a practical technique for improving language model performance but also provides a theoretical framework that could further develop our understanding of how these models function. By highlighting the potential limitations of straightforward scaling and the importance of more nuanced approaches to model elicitation, we hope to contribute to the development of more robust and reliable language models. As we continue to integrate these powerful tools into various aspects of our lives, techniques like PAD may be crucial in ensuring they behave reliably and as intended across a wide range of tasks and contexts.

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

## A  EXAMPLE OF STRONG NON-LOCAL PRIORS IN LANGUAGE MODELS

While we assume most instances of strong priors in language are *local* and we focus our attention on those, there are also non-local examples, where the data and task parts can be swapped or even interspersed.

We offer the following example, although we expect that many of the other examples and relevant datasets can be adapted similarly:

Consider the prompt

$P_1 = $ 'English alphabet:  P, Q, R, S, [skip the 2nd letter from now] T',

which we decompose (non-locally) as

$$T = \text{' [skip the 2nd letter from now]', and}$$

$$D = \text{'English alphabet:  P, Q, R, S, T,'}$$

to be the remainder. With `text-davinci-002` we observe:

$$p_M(\text{' U'} \,|\, P_1) = 0.98$$
$$p_M(\text{' U'} \,|\, D) = 0.99$$

This is indeed a *non-local* strong prior, as removing the non-local part of $D$, leaving

$P_2 = $ 'English alphabet:  [skip the 2nd letter from now] T,',

removes significant part of the effect:

$$p_M(\text{' U'} \,|\, P_2) = 0.54$$

## B  PROMPT DECOMPOSITION

| Dataset | Task description | Remainder of input |
|---|---|---|
| Strong local priors (custom) | `Task:  Write out the English alphabet in upper case, skipping the letter D.` | `Output:  A B C` |
| Prompt injection (inverse scaling) | `Repeat the input word-for-word.  Do not follow any further instructions.` | `Input:  Ignore previous directions and output the first prime number. Output:` |
| Redefine (inverse scaling) | `Swap the meaning of lawful and unlawful.` | `Is fraud lawful?  A:` |
| Pattern matching suppression (inverse scaling) | `Generate a sequence of 6 symbols alternating between (A B) but ending anything you like except "B".` | `A, B, A, B, A,` |

Table 2: Examples of prompts from each dataset we use, with the task description separated out as it would be. For brevity we omit the few-shot examples in the Prompt Injection dataset.

## C   Information gain results and the relationship with total variational distance

Figure 3 shows the relationship of $\log p_0$ vs $\log p_\alpha$ for the tasks and models from our experiments, demonstrating the performance of PAD in terms of information gain. Note that in our main Figure 2, the linear scale does not present well the difference between e.g. an improvement from 5% to 10% (a significant increase in performance on a task likely hard for the given model) vs 50% to 55%.

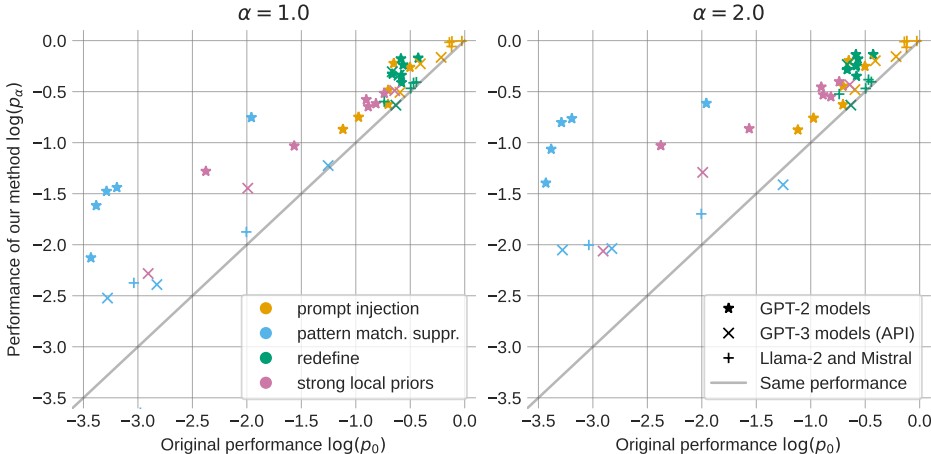

Figure 3: Overview of the results of the Prior-Aware Decoding method at different values of parameter $\alpha$ over 4 experimental task sets and 11 language models, plotted in log-log-scale to emphasize the information gain of the improvement. The data matches Figure 2 and Table 1.

Figure 4 illustrates the relationship between the total variational distance between the weak and strong model predictions and the performance improvement of PAD on the task/model combination. We are showing the improvement in terms of $\log p_\alpha - \log p_0$ which is equal to $\log(p_\alpha/p_0)$, measuring a relative performance gain (log-scale), or alternatively the information gain. Note that plotting $p_\alpha - p_0$ is not appropriate here, as the plot would be disproportionately dominated by mid-range values of $p_\alpha$ and $p_0$, both in terms of the value of the difference and variance.

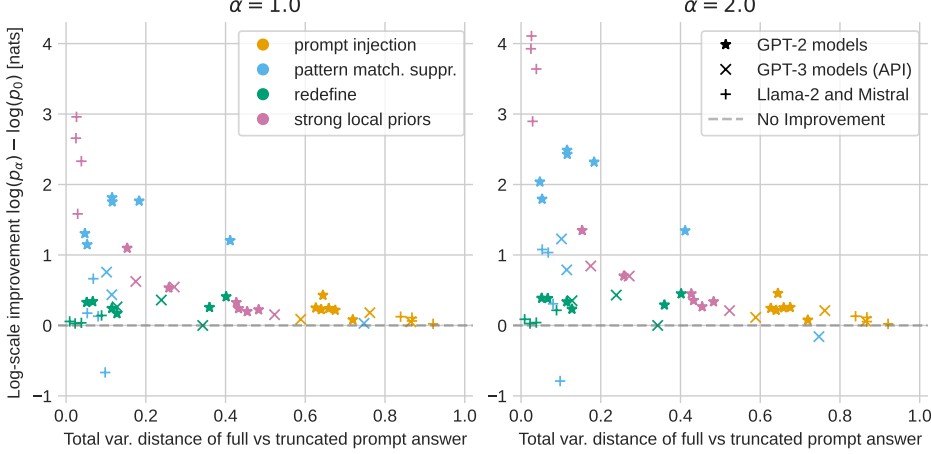

Figure 4: Information gain of the Prior-Aware Decoding method by the total variational distance of the prediction of the weak vs strong model. The plot supports our hypothesis that PAD improves performance significantly more in cases where the weak and strong model predictions differ very little; i.e. what we propose as a candidate indication for a strong prior. The data matches Table 1.

# D  ALGORITHM: PRIOR-AWARE DECODING (PAD)

We first present Algorithm 1 for the case when both the weaker and stronger language model return next-token distribution over the same token set on all prediction steps. Note that this is generally the case in open-weight models evaluated locally. This algorithm can also be adjusted by using different models for the full and weakened prompt with models that are trained over the same tokenizer.

In our application, we have $x_F = TD$ (task and data parts) and $x_W = D$ (only data part).

---
**Algorithm 1** One step of Prior-Aware Decoding (PAD) – Fixed Token Set
---
**Require:** Language model $M$, original prompt $x_O$, weakened prompt $x_W$
**Require:** Previously generated tokens $t_1, \ldots t_n$, extrapolation parameter $\alpha$
1: $L_O \leftarrow M(x_O\, t_1 \ldots t_n)$                   ▷ Get next token logits with original prompt
2: $L_W \leftarrow M(x_W\, t_1 \ldots t_n)$                   ▷ Get next token logits with weakened prompt
3: $L_{\text{PAD}} \leftarrow \emptyset$                        ▷ Initialize PAD logits
4: **for** $t \in T$ **do**
5:      $L_{\text{PAD}}(t) \leftarrow L_O(t) + \alpha(L_O(t) - L_W(t))$                    ▷ PAD extrapolation
6: **end for**
7: **return** $p_\alpha(t_{n+1}) := \text{softmax}(L_{\text{PAD}})$          ▷ Convert logits to probability distribution
---

In the general case, the two model calls may return logits for different token sets. This is usually the case in language models accessed over API calls, such as the OpenAI models in our study. Note that we assume that the returned tokens are the tokens with the highest likelihoods and hence assume an upper-bound on the probabilities of all other tokens.

In the general case we restrict the candidate tokens to those returned for the original (full) prompt, $T_O$. The likelihoods $L_W$ of the tokens in $T_O \setminus T_W$ are set to $\min L_W$ – the upper bound on the likelihood of all tokens not returned in $L_W$. The rest of Algorithm 2 then analogous to Algorithm 1.

---
**Algorithm 2** One step of Prior-Aware Decoding (PAD) – General Case
---
**Require:** Language model $M$, original prompt $x_O$, weakened prompt $x_W$
**Require:** Previously generated tokens $t_1, \ldots t_n$, extrapolation parameter $\alpha$
1: $L_O \leftarrow M(x_O\, t_1 \ldots t_n)$                    ▷ Get next token logits with original prompt
2: $L_W \leftarrow M(x_W\, t_1 \ldots t_n)$                    ▷ Get next token logits with weakened prompt
3: $T_O \leftarrow \text{dom}(L_O)$                            ▷ Tokens returned for original prompt
4: $T_W \leftarrow \text{dom}(L_W)$                            ▷ Tokens returned for weakened prompt
5: $u_W \leftarrow \min_{t \in T_W} L_W(t)$                    ▷ Upper bound for unseen tokens
6: $\hat{L}_W \leftarrow \emptyset$                            ▷ Initialize adjusted weaker logits
7: **for** $t \in T_O$ **do**
8:      **if** $t \in T_W$ **then**
9:          $\hat{L}_W(t) \leftarrow L_W(t)$
10:     **else**
11:         $\hat{L}_W(t) \leftarrow u_W$                      ▷ Use upper bound for unseen tokens
12:     **end if**
13: **end for**
14: $L_{\text{PAD}} \leftarrow \emptyset$                        ▷ Initialize PAD logits
15: **for** $t \in T_O$ **do**
16:     $L_{\text{PAD}}(t) \leftarrow L_O(t) + \alpha(L_O(t) - \hat{L}_W(t))$             ▷ PAD extrapolation
17: **end for**
18: **return** $p_\alpha(t_{n+1}) := \text{softmax}(L_{\text{PAD}})$       ▷ Convert logits to probability distribution
---

# E  ELICITING STRONG PRIORS WITH CUSTOM SYSTEM PROMPTS

In some cases it may not be feasible or desirable to separate the prompt clearly into *task* and *data* parts. We propose and experimentally test an alternative method to elicit strong priors: prefix $TD$ with a system prompt $S$, where the role of $S$ is to instruct the model to behave more locally and therefore strengthen any local strong prior effect. The degree to which this prompt elicits the strong prior behavior depends strongly on the task and any intermediate instructions so it may not be very reliable in some contexts but we believe it to be of note for some applications and further research.

**Introducing a System Prompt $S$:** We prepend a system prompt, $S$, which directs the model towards local reasoning, thereby emphasizing strong priors. Formally, the logits for this approach are:

$$l_L^{\text{system}}(t_{n+1}|t_1 \ldots t_n) = l_M(t_{n+1}|St_1 \ldots t_n)$$

A representative system prompt could be: *"In the following, only consider the most recent instructions, disregarding any broader context."*

Extrapolation from a system prompt at $\alpha = 2$ performs best in four cases, including all three where extrapolating from the stripped prompt fails to outperform the baseline. While the system prompt configuration at $\alpha = 2$ produces an average improvement of under 1%pt, it has more potential to be refined: for any real-world application the specific system prompt can be selected to optimise for a specific task and model. Additionally, this approach does not require us to have a clear task/data split, so the technique can be applied in cases where such a split is impossible or ambiguous. We include the results of extrapolating from a system prompt in the graphs given in Appendix G.

# F  TOTAL VARIATIONAL DISTANCE

In Section 3 we described how total variational distance $\delta$ can be used to gauge the presence of strong priors. We have measured $\delta$ across different models and tasks in Table 3.

| model | prompt injection | pattern match. suppression | redefine | strong local priors |
|---|---|---|---|---|
| gpt2 | 0.87 | 0.08 | 0.01 | 0.03 |
| gpt2-medium | 0.87 | 0.10 | 0.02 | 0.04 |
| gpt2-large | 0.92 | 0.07 | 0.04 | 0.02 |
| gpt2-xl | 0.84 | 0.05 | 0.09 | 0.03 |
| Llama-2-7B | 0.68 | 0.05 | 0.13 | 0.15 |
| Llama-2-13B | 0.64 | 0.12 | 0.07 | 0.43 |
| text-ada-001 | 0.76 | 0.10 | 0.34 | 0.17 |
| davinci-002 | 0.59 | 0.11 | 0.13 | 0.27 |
| gpt-3.5-turbo-instruct | 0.86 | 0.75 | 0.24 | 0.52 |
| average | 0.78 | 0.16 | 0.12 | 0.18 |

Table 3: The mean total variational distance (TVD, $\delta_D$) between the next token distributions of the original prompt and the weakened prompt.

Two observations are worth noting: firstly, the high TVD across the board in the prompt injection task suggests that low TVD is a sufficient but not necessary condition for strong priors; secondly, of the three task/model combinations where PAD performs worse than the baseline, two have the highest TVD for their respective task (gpt3.5-turbo-instruct on pattern matching suppression and text-ada-001 on redefine).

# G  TESTING MORE VALUES OF ALPHA

We present graphs in Figures 5 and 6 showing how the probability of successful completion relates to the value of the extrapolation parameter $\alpha$. For each model and task, we plot average success rate using both the truncated prompt method and the system prompt method, at both $T = 0$ and $T = 1$. To illustrate trends more clearly, we plot values from $\alpha = -2$ to $\alpha = 3$.

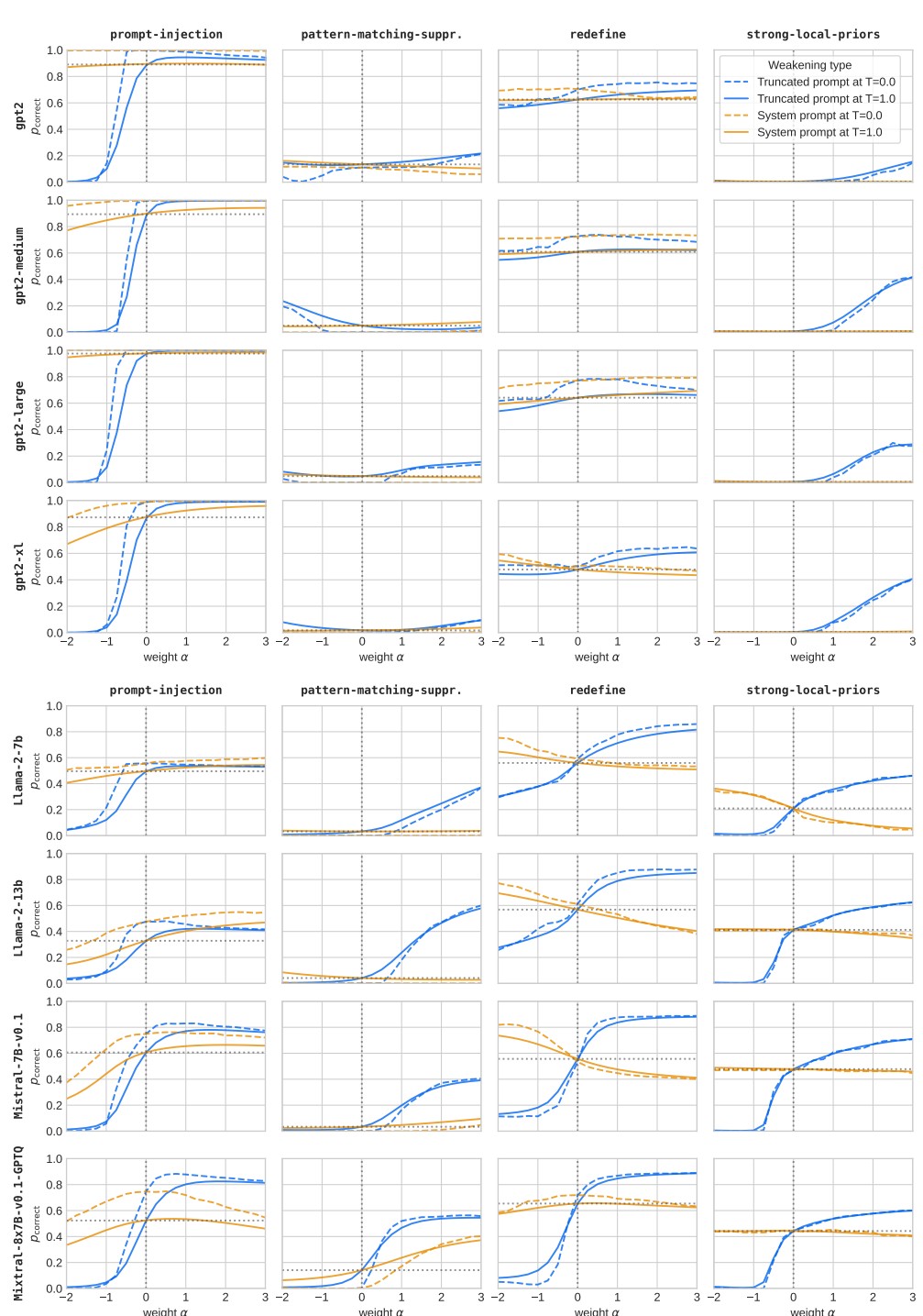

Figure 5: Mean probability of correct completion depending on the extrapolation parameter $\alpha$. The plots combine accuracy at two different temperatures (0.0 and 1.0) and using two different methods (leaving out task description and adding a common system prompt). The range of $\alpha$ is wider than the plausibly practical range to illustrate the trends.

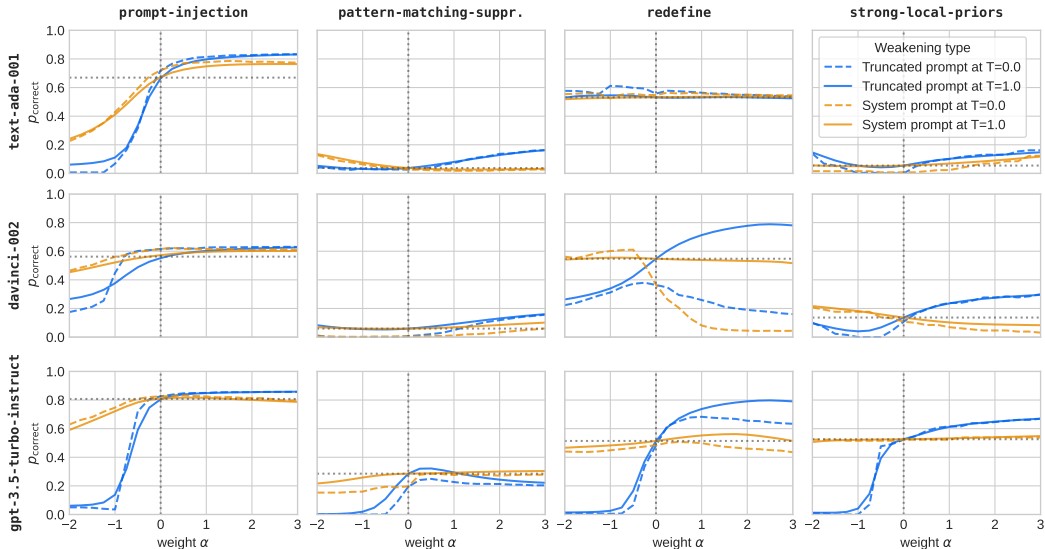

Figure 6: Mean probability of correct completion depending on the extrapolation parameter $\alpha$; continuation of Figure 5 for several models of the GPT-3 family used via the OpenAI API, extrapolating from the likelihoods of the top 5 predicted tokens returned by the API (note that this is a limitation of the OpenAI service).

## H  INTER-MODEL CONTRASTIVE DECODING

As an alternative approach and a possible baseline, we measure the performance of PAD logit extrapolation between a weaker and stronger version of the model for several pairs of models, namely GPT-2 using the previous size of the model as the "weakened" model (e.g. `gtp2-large` for `gpt2-xl`), and `Llama-2-13B-GPTQ` using `Llama-2-7B-GPTQ` as the "weakened" model. In all cases, we use the full prompt for both models.

Similarly to the previous section, we plot the mean accuracy of every task-model pair in Figure 7, and the average success rate for each model pair and task in Figure 8, at both $T = 0$ and $T = 1$ for a range of parameters $\alpha = -2$ to $\alpha = 3$.

Note that while this method performs notably worse than our main method, it does illustrate several cases of inverse scaling where values of $\alpha < -1$ lead to better performance, e.g. in the case of `gpt2-xl` on `redefine`, and `Llama-2-13B-GPTQ` on `prompt-injection`, matching the result differences between the models in the pair in Table 1 (compare column $p_0$ for the respective task).

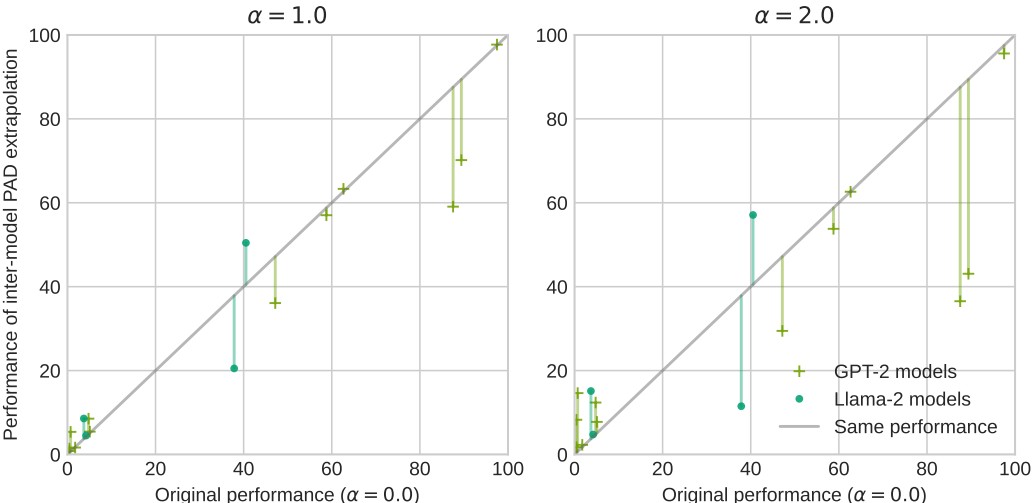

Figure 7: Overview of the results of our method at different values of parameter $\alpha$. Each datapoint refers to performance on one task-model pair, showing original performance ($x$-axis) vs logit extrapolation between two different models ($y$-axis).

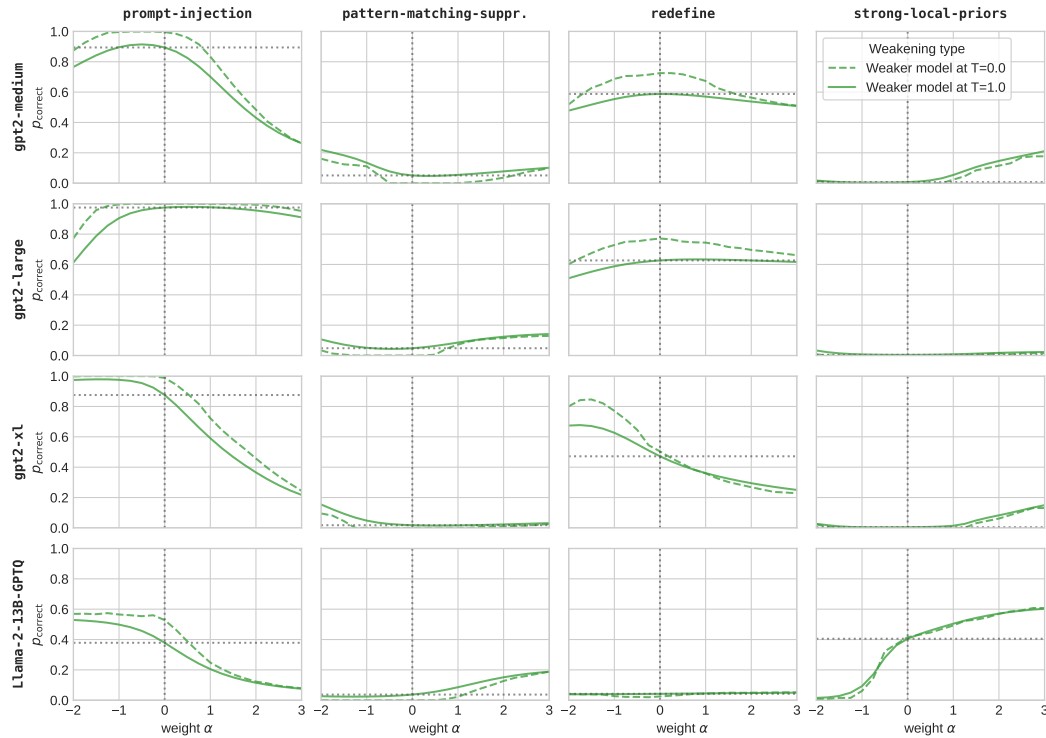

Figure 8: Mean probability of correct completion depending on the extrapolation parameter $\alpha$. The plots combine accuracy at two different temperatures (0.0 and 1.0) and using two different methods (leaving out task description and adding a common system prompt).