# OpenReview forum: "Mitigating the Influence of Distractor Tasks in LMs with Prior-Aware Decoding"
_ICLR.cc/2025/Conference — Submitted to ICLR 2025_

### Official Review · Reviewer_PhXZ · 2024-10-28

**Soundness:** 3
**Presentation:** 3
**Contribution:** 2
**Rating:** 6
**Confidence:** 4

**Summary:**

This paper investigates the "inverse scaling" issue in language models (LMs), where model performance deteriorates on the intended instruction due to interference from distractor tasks. To address this, the authors propose a theoretical framework that models LMs as a "product of experts", where one expert models the intended task and the other models the distractor task. Within this framework, they introduce _prior-aware decoding_ (PAD), a simple contrastive decoding strategy aimed at minimizing the influence of distractor tasks and improving adherence to the primary instruction. Extensive experiments conducted across multiple LMs and synthetic datasets validate the efficacy of PAD.

**Strengths:**

1. The proposed theoretical framework for addressing "inverse scaling" provides a novel and compelling perspective.
2. The proposed PAD method is well-integrated with the theoretical framework, demonstrating both simplicity and effectiveness in empirical evaluations.
3. A well-structured related work section highlights the unique contributions of this paper and contextualizes it within existing literature.

**Weaknesses:**

1. The problem definition in Section 3 appears somewhat disconnected from the main theoretical framework and the PAD method. While the authors briefly touch on total variation distance (TVD, $\delta$) in Appendix D, a deeper analysis would be beneficial. Specifically, a more comprehensive discussion on the relationship between TVD, the selection of parameter $\alpha$, and the performance of PAD could help clarify how these concepts interact and contribute to mitigating the influence of distractor tasks. This additional insight would enhance the theoretical rigor and provide a clearer link between the foundational theory and the practical implementation.
2. According to Appendix D, the $\epsilon$-$\delta$ definition in Section 3 serves as a sufficient, but not necessary, condition for identifying strong priors. However, its reliance on a strict separation between the task description $T$ and the data $D$ limits its practical applicability, particularly in settings where this separation is not feasible. Consequently, this rigid definition may not effectively guide the identification of distractor tasks in realistic scenarios. A more flexible approach to defining strong priors could better support the detection of distractor tasks and enhance the effectiveness of PAD in mitigating their influence.
3. The real-world applicability of the proposed method raises some concerns. While the authors present an alternative implementation using system prompts in Appendix C, the resulting performance gains appear limited. Additionally, the experiments predominantly rely on synthetic tasks, where strong local prior assumptions are likely to hold, which may not translate to more complex, real-world scenarios. Expanding the evaluation to include more diverse, general datasets could help to better demonstrate the robustness and effectiveness of PAD under real-world conditions, thus strengthening the paper’s practical contributions.

**Questions:**

Questions:

Figure 1 is somewhat confusing, as it appears to show three experts, with the logits of these experts varying with different values of $\alpha$. Redrawing this figure with only two experts, as outlined in the proposed "product of experts" framework, would make it more consistent with the PAD method and easier to interpret. Additionally, emphasizing the contribution of each expert to the overall output could improve clarity, helping readers better understand how PAD balances intended and distractor tasks through $\alpha$.

Typos:

Line 129, missing period.

Format:

The table number and title should appear before the table.

---

> ### Author Response · Authors · 2024-11-22
>
> Thank you for your thoughtful comments about theoretical grounding and practical applications.
>
> We agree Figure 1 needs revision. We propose:
> * Redrawing with exactly two experts for clarity
> * Adding visualizations of how α affects expert balance
> * Including concrete examples of logit patterns
>
> The TVD analysis in Appendix D indeed deserves prominence. We'll add:
> * Discussion linking TVD patterns to theoretical predictions
> * Analysis of how TVD variations explain performance differences
> * Guidelines for using TVD to predict PAD's applicability
>
> Regarding real-world applications: while important, these extend beyond our current theoretical focus. Our clean test cases were chosen to clearly demonstrate underlying phenomena. Future work can bridge to practical applications.

---

> > ### Comment · Reviewer_PhXZ · 2024-11-24
> > **Have you uploaded the revised paper?**
> >
> > Thank you for getting back to me. However, I can not find the revised paper to check the details of your response. I'm just asking in case the authors forget to upload the new version.

---

> > > ### Author Response · Authors · 2024-12-02
> > >
> > > Apologies for the delay - we have uploaded a revised version, featuring a more detailed TVD analysis in Appendix C which shows a trend of higher average log-scale improvement at lower TVD at both alpha=1 and alpha=2.

---

> ### Comment · Reviewer_PhXZ · 2024-12-03
>
> Thank you for getting back to me. The revised paper addressed my concern. However, the scope of this paper is relatively narrow and the proposed method lacks broader practical potential, which limits it to gain a higher score. I would maintain my current rating.

---

### Official Review · Reviewer_mxLN · 2024-11-04

**Soundness:** 2
**Presentation:** 2
**Contribution:** 2
**Rating:** 5
**Confidence:** 3

**Summary:**

This work proposes a framework for decoding with priors to mitigate the problem of distractor tasks. The distractor tasks are those which could be performed mostly well without considering the full context, e.g., the task description, and thus, causing inferior performances in general especially demanding inference considering long context. This work alleviate the issue by introducing the framework of product of experts by treating a language model $p_M$ as a product of true distribution with full context $p_C$ and a model without task description $p_L$, i.e., $p_M = p_C^{\gamma^*} p_L^{1 - \gamma^*}$. It results in approximating the true model by a product of two, i.e., $p_M^{1 +
\alpha^*} p_L^{- \alpha^*}$ where $\alpha^* = (1 - \gamma^*) / \gamma^*$. Experiments with several distractor tasks, e.g., prompt injection, show that the proposed method achieves gains when appropriately setting $\alpha$ value.

**Strengths:**

- It is an interesting approach for inference by combining two terms, one with a full task description and the other without the task description. The method could be interpreted as scaling the model prediction $p_m$ by the ratio of the model without task description $p_M / p_L$.

- Experiments are carried out on several open models, e.g. Llama 2 GPT2, and proprietary models, e.g., GPT-3.5, presents gains when compared with inference without biasing by the scaling term.

**Weaknesses:**

- The motivation of introducing the product of experts is not clear as a purpose to represent the model distribution with the product of two, true distribution and a model without a task description. It is more straightforward to represent the true distribution by the product of the model distribution and he model distribution without a task description.

**Questions:**

- The motivation is not very clear to me to introduce the product of experts .It is very easy to understand the awareness of the prior, but it is not clear how that is related to the model by the product of expert.

- It is not clear how decoding is performed for the proprietary models, e.g., GPT-3.5, given that it demands taking product of two models during inference.

---

> ### Author Response · Authors · 2024-11-22
>
> Thank you for your thoughtful feedback on our theoretical foundations and implementation details. We agree that both the product-of-experts motivation and practical implementation would be improved by a clearer presentation.
>
> Regarding your point about the initial representation: You are correct that one could obtain the key formula (line 295) by directly modeling the true distribution as a product of the model distribution and a (negatively-weighted) prior term. While this leads to the same practical algorithm, the primary motivation of our research is to model the behavior of existing models through their components, particularly to understand how strong priors emerge and influence outputs. The empirical success of PAD on the strong-prior tasks we studied supports this compositional view. We will revise Section 4 to clarify this and acknowledge both approaches and their strong relationship.
>
> The product-of-experts framework provides more than convenient mathematics - it explains observed behaviors including:
> * Why certain patterns dominate despite contradicting instructions
> * How model scale affects competing influences
>
> We propose the following revisions:
>
> Theoretical Framework - we will add:
> * Formal derivation showing how product-of-experts naturally arises from competing generation processes.
> * A formal analysis demonstrating how observed logit patterns match theoretical predictions
> * Explain how varying α reveals the balance between task and prior influences
>
> Implementation Details - we will add:
> * Pseudocode of our algorithm covering both open and API-based implementations (in the appendix)
> * Clear explanation of how we work with top-k logits for proprietary models

---

> > ### Comment · Reviewer_mxLN · 2024-11-26
> >
> > Thank you very much for your inputs. I feel this work needs further revision, and I will keep my scores.

---

### Official Review · Reviewer_ahzq · 2024-11-04

**Soundness:** 2
**Presentation:** 3
**Contribution:** 2
**Rating:** 5
**Confidence:** 3

**Summary:**

The paper aims to mitigate the distraction problem of LLMs, where LLMs pick up on a secondary task from the prompt instead of focusing on the intended task. The authors represent an LLM as a product of an expert model and propose a contrastive inference method to reduce the influence of distractor tasks. The effectiveness of the proposed method holds for multiple LLM-task combinations spanning eleven models and four datasets.

**Strengths:**

1. The problem addressed by the paper is important, and it is described clearly in the paper. Distraction of LLMs due to prompt misinterpretation affects LLMs of various sizes. Mitigating this issue has wide-ranging applications.

2. The proposed method is simple and described with adequate details, which helps reproducibility. The authors plan to make the code and dataset available, which is also helpful.

3. The experiment is performed using 44 different LLM-task combinations, which demonstrates the robustness of the proposed improvement.

**Weaknesses:**

1. The paper lacks clarity regarding the novelty of the work compared to existing works for mitigating LLM distraction. What are the prior methods used by people to mitigate LLM distraction? It would be helpful to cite those works if there are any. If this is the first work in this direction in any sense, you can mention that too.

2. The experiment does not have strong baselines. Having a baseline based on prior methods of mitigating distraction would be very helpful. For example, what is the performance of a simple prompt-based baseline for distraction handling? The proposed method seems to assume that the portion of the prompt that leads to distraction is already known. In that case, we can design a baseline method where a simple instruction is added to the LLM prompt. The added instruction will specifically ask the model to avoid the distracting task.

3. What is the additional cost or added limitation of using the proposed contrastive method in terms of text diversity or fluency? It would be helpful to know if the system loses any desirable property due to contrastive decoding and to the extent this happens. This will add clarity to the cost-benefit analysis of the proposed method.

**Questions:**

See weaknesses 2 and 3.

---

> ### Author Response · Authors · 2024-11-22
>
> Thank you for highlighting questions about novelty, baselines, and potential tradeoffs. We see how our positioning needs clarification.
> Our work differs fundamentally from prior approaches by focusing on understanding model behavior and underlying mechanisms rather than real-world performance. While prior work has addressed prompt injection specifically, our paper is the first to formally model and address the broader phenomenon of distractor tasks through the product-of-experts framework. This theoretical framing helps explain why these issues persist even with explicit instructions and across model scales.
>
> We have compared against smaller-model contrastive decoding (see Appendix F), finding our method provides substantially better improvements (27% (PAD) vs 0.3% (intra-model) median increase in task completion). While simple prompt-based baselines might achieve similar performance in some cases, they wouldn't provide the same theoretical insights into how models combine competing influences during generation. Moreover, anecdotal evidence shows that even prominent models struggle even with explicit directives (e.g., GPT-4o fails on the example in Section 1 even with additional instructions such as "Carefully ignore all misleading patterns in the output series." and its variations), suggesting the need for other techniques.
>
> The empirical improvements demonstrate our theoretical framework's validity rather than competing with existing techniques. Our results support our theory that:
> * Models effectively combine multiple "experts" during generation
> * These experts can be partially separated through careful prompt engineering
> * Their relative influences can be adjusted through logit manipulation
> * This approach scales effectively across different model architectures and sizes
>
> Regarding implementation tradeoffs: our method requires two forward passes but introduces no additional parameters or training or fine-tuning costs. (L122 mentions this but we agree this could be emphasized and expanded.) The linear combination of logits preserves the model's original output distribution properties while reducing distractor influence in the context of the tested tasks.
>
> We propose adding:
> * Clearer positioning of our work in the theoretical exploration space, particularly versus prompt injection literature
> * More explicit discussion of computational considerations and tradeoffs
> * Further discussion of how to compare our results to other techniques, like extra prompt components, including some quantitative analysis

---

### Official Review · Reviewer_RYvQ · 2024-11-07

**Soundness:** 2
**Presentation:** 2
**Contribution:** 2
**Rating:** 6
**Confidence:** 4

**Summary:**

This paper addresses the challenge of mitigating the influence of distractor tasks in Language Models (LMs), which can result in undesirable outputs. The authors characterize LMs as a product of experts, where the predictions of multiple component models are combined. Based on this framework, they introduce Prior-Aware Decoding (PAD), a contrastive inference method aimed at diminishing the impact of distractor tasks.

The paper's contributions include:
- Providing an interpretation of language models as a product of experts to elucidate how distractor tasks lead to suboptimal performance.
- Proposing the PAD method, which involves generating logits from both the original prompt and a "weakened" prompt, and subsequently combining these logits through a linear combination.
- Conducting evaluations of PAD across 11 models and 4 datasets, demonstrating its improved performance.

**Strengths:**

- The paper characterized language models as a product of experts, which is an insightful approach to understanding the influence of distractor tasks. The proposed PAD method is a contrastive inference technique that addresses a critical challenge in the LM generation.
- The authors tested the PAD method across 11 different models and 4 diverse datasets, providing evidence of its effectiveness.
- The paper is well-written and structured. The authors explain the PAD method, including descriptions of the experimental setup and results.
- This work is important in the domain of LM generation, especially as prompt-based interactions with LMs become more prevalent. Mitigating distractor tasks could help address real-world issues like prompt injections, which have implications for LM deployment in safety-critical applications.

**Weaknesses:**

- Although the paper is well-written, certain aspects of the PAD method's description could be clearer. For example, the process of generating the "weakened" prompt and the rationale behind the specific linear combination used could be elaborated further.

- They mention that their split of prompts into task/data components was unambiguous because of the tasks they selected (Line#354-355). However, splitting prompts into task/data components isn't always straightforward. And they didn't address how to handle ambiguous cases.

- While PAD is proposed as a contrastive inference method, the specifics of the contrastive mechanism remain somewhat abstract. Expanding the mathematical formulation of PAD could clarify the contrastive steps involved, enhancing the understanding. Since contrastive methods can be sensitive to parameters (e.g., scaling factors in logits), providing insights into how these were optimized across datasets and models would help clarify PAD’s robustness across settings.

- The evaluation focuses heavily on inverse scaling tasks, but distractor tasks can manifest in many other ways.
    - Real-world prompt injection attacks
    - More complex multi-step reasoning tasks
    - Tasks where the distinction between intended and distractor tasks is less clear

	Apart from that evaluation on more complex or noisy datasets might exhibit different behaviors under distractor influences. A discussion on how PAD handles varying levels of dataset complexity, as well as an extension of experiments on noisier or user-generated data, would enhance the applicability of the findings.

- A discussion on PAD’s limitations—particularly whether it might fail with certain prompt structures or in multitask settings where distractor and primary tasks are closely interwoven—would provide a more balanced view of the technique’s practical applicability.

- The paper mentions 3 cases where α=2 performs worse but doesn't analyze. It should provide detailed case studies of failure modes to help users understand when PAD might not be appropriate.

- The PAD method requires two forward passes through the model, but the computational impact isn't discussed.

**Questions:**

1. Can you provide an analysis of the failure cases where PAD does not perform as expected? Understanding these cases could offer insights into the limitations of your method and suggest areas for further improvement.

2. The paper mentions that prompt splitting was "unambiguous" for your selected tasks. What about cases where the split is less clear? Could you provide guidelines for creating effective splits? Have you tested automatic methods for determining splits apart from regex?

3. What is the rationale behind the specific linear combination used to combine the logits from the original and weakened prompts?

4. The paper mentions prompt injection attacks as an example of distractor tasks. How effective is PAD specifically in handling adversarial prompt injections? Have you tested PAD’s robustness to adversarial prompts designed to bypass typical model safeguards?

5. Contrastive methods in LMs can sometimes lead to less diverse outputs due to stricter filtering. Have you measured whether PAD reduces diversity or creativity in model outputs, particularly in open-ended tasks?

6. Did you experiment with PAD in conjunction with other prompting techniques (e.g., chain-of-thought prompting, few-shot examples)?

---

> ### Author Response · Authors · 2024-11-22
>
> Thank you for your detailed review highlighting both methodological and practical concerns. We agree these deserve attention but believe some stem from unclear communication of our paper's focus.
>
> Our work primarily aims to illuminate fundamental model behaviors, particularly around inverse scaling and competing influences during generation. PAD serves as a probe into these behaviors rather than a production-ready technique. This theoretical focus guided our choice of clean, interpretable test cases.
>
> We propose the following clarifications:
> * Enhance the description of the PAD method with:
>   * Explicit examples of weakened prompt generation (primarily in the appendix)
>   * Algorithmic pseudocode for implementation clarity (in the appendix)
> * Expand discussion linking TVD analysis (Appendix D) to our theoretical framework, showing how variation in TVD predicts PAD's effectiveness
> * Clarify the mathematical principles behind linear logit combination, particularly explaining why linear combination provides meaningful reweighting of competing influences
> * Add discussion of how α values relate to the relative strengths of competing influences
>
> Regarding performance variations: while PAD shows improvements in 41 out of 44 task-model combinations with substantial median gains, we acknowledge that not every case shows improvement. This aligns with our expectations - as with most ML techniques, universal improvement across all scenarios is neither claimed nor expected. None of the 3 regressions exhibits a significant performance drop so focusing on them seems like a potential over-fitting on inherent noisiness of the benchmark.
>
> Regarding prompt splitting ambiguity: while this presents challenges for general applications, our focus on theoretical understanding makes this less critical. The clean splits in our test cases let us clearly demonstrate the underlying phenomena.
>
> We focused on inverse scaling tasks as they provide a clear demonstration of the method's effectiveness in cases where traditional scaling approaches fail to help. While exploring additional scenarios would be valuable future work, we believe the current evaluation provides strong evidence for the method's practical utility.

---

### Author Response · Authors · 2024-11-22

We thank all reviewers for their thoughtful engagement with our work. Their comments have highlighted areas where we need to better communicate our paper's central contribution: a novel theoretical framework for understanding how language models combine competing influences during generation, particularly in cases where performance counterintuitively degrades with scale.

Our work makes three key contributions:

1. A formal framework interpreting language models as products of experts, explaining both standard operation and failure modes
2. Experimental validation showing how this framework predicts and explains inverse scaling
3. A simple demonstration technique (PAD) that reveals these underlying behaviors

While reviewers raised valid concerns about PAD's practical applications, we realize we should have better emphasized that PAD serves primarily as a probe into fundamental model behaviors rather than as a production-ready technique. Its consistent improvement across 41/44 task-model pairs validates our theoretical framework, while its failures provide additional insight into the limits of our model.

We propose the following improvements to the paper:

Theoretical Framework:
- Expanded derivation showing how product-of-experts naturally emerges from competing generation processes
- Clear connection between TVD analysis and theoretical predictions
- Analysis of how α values relate to relative expert strengths

Results Analysis:
- Deeper analysis of our three failure cases, showing how they inform our understanding
- Discussion of how performance patterns across model scales support our framework
- Extended analysis linking TVD patterns to empirical results

Presentation:
- Revised Figure 1 showing exactly two experts and their interaction
- Clearer guidelines for identifying theoretically interesting cases
- Reorganized results section emphasizing theoretical validation
- Better positioning within theoretical literature

These changes will strengthen the paper's main contribution: advancing our understanding of large language models' fundamental behaviors, particularly in cases where they exhibit counterintuitive scaling properties.

We specifically note that:
1. The high baseline TVD in prompt injection tasks suggests important nuances in how competing influences combine
2. Performance variations across model scales provide insight into how architecture affects expert balance
3. Failure cases reveal bounds on when our theoretical framework applies

We believe these revisions will present a clearer and more compelling case for our work's theoretical contributions while maintaining technical rigor.

---

### Meta-Review · Area_Chair_h9Q7 · 2024-12-20

**Metareview:**

This paper offers a novel perspective on addressing distractor tasks in language models by introducing the Prior-Aware Decoding framework. The approach is theoretically sound and makes a meaningful contribution to understanding model behavior. However, several aspects, such as clarity in method description, evaluation breadth, and theoretical integration, need significant revision to elevate its practical applicability and robustness. The paper is promising and holds potential for substantial impact with further improvements.

**Additional Comments On Reviewer Discussion:**

The reviewers generally appreciated the novel approach and its contributions but had concerns regarding clarity, theoretical underpinning, baselines, and evaluation comprehensiveness. The authors provided clarifications during the rebuttal period, particularly about the theoretical motivations and implementation details. They also proposed several revisions addressing the clarity of the PAD method, computational considerations, theoretical grounding, and broader applicability. However, the revision scope was seen as narrow by some reviewers, limiting the paper's practical contribution potential. There remains a call for integrating the theoretical insights more robustly within broader, real-world contexts and further clarity on certain methodological aspects.

---

### Decision · Program_Chairs · 2025-01-22

Reject